# Intraoperative Assessment of Fluid Responsiveness in Normotensive Dogs under Isoflurane Anaesthesia

**DOI:** 10.3390/vetsci8020026

**Published:** 2021-02-11

**Authors:** Despoina Skouropoulou, Luca Lacitignola, Caterina Di Bella, Marzia Stabile, Claudia Acquafredda, Nicola Brienza, Salvatore Grasso, Antonio Crovace, Fabrizio Iarussi, Francesco Staffieri

**Affiliations:** 1Section of Veterinary Clinics and Animal Production, Department of Emergency and Organ Transplantation, Aldo Moro University, 70010 Bari, Italy; semeli.amenta@gmail.com (D.S.); luca.lacitignola@uniba.it (L.L.); caterina.dibella@uniba.it (C.D.B.); marzia.stabile@uniba.it (M.S.); claudia.acquafredda@uniba.it (C.A.); antonio.crovace@uniba.it (A.C.); fabrizio.iarussi@uniba.it (F.I.); 2Ph.D. Course in Organs and Tissues Transplantation and Cellular Therapies, Department of Emergency and Organ Transplantations, Aldo Moro University, 70124 Bari, Italy; 3Section of Anesthesia and Intensive Care, Department of Emergency and Organ Transplantation, Aldo Moro University, 70124 Bari, Italy; nicola.brienza@uniba.it (N.B.); salvatore.grasso@uniba.it (S.G.)

**Keywords:** dogs, fluid responsiveness, pulse pressure variation, systolic pressure variation, stroke volume variation, plethysmographic variability index

## Abstract

The aim of this study was to evaluate the incidence of fluid responsiveness (FR) to a fluid challenge (FC) in normotensive dogs under anaesthesia. The accuracy of pulse pressure variation (PPV), systolic pressure variation (SPV), stroke volume variation (SVV), and plethysmographic variability index (PVI) for predicting FR was also evaluated. Dogs were anaesthetised with methadone, propofol, and inhaled isoflurane in oxygen, under volume-controlled mechanical ventilation. FC was performed by the administration of 5 mL/kg of Ringer’s lactate within 5 min. Cardiac index (CI; L/min/m^2^), PPV, (%), SVV (%), SPV (%), and PVI (%) were registered before and after FC. Data were analysed with ANOVA and ROC tests (*p* < 0.05). Fluid responsiveness was defined as 15% increase in CI. Eighty dogs completed the study. Fifty (62.5%) were responders and 30 (37.5%) were nonresponders. The PPV, PVI, SPV, and SVV cut-off values (AUC, *p*) for discriminating responders from nonresponders were PPV >13.8% (0.979, <0.001), PVI >14% (0.956, <0.001), SPV >4.1% (0.793, <0.001), and SVV >14.7% (0.729, <0.001), respectively. Up to 62.5% of normotensive dogs under inhalant anaesthesia may be fluid responders. PPV and PVI have better diagnostic accuracy to predict FR, compared to SPV and SVV.

## 1. Introduction

Perioperative haemodynamic stability is important for optimizing oxygen delivery and tissue perfusion and reducing postoperative complications [1]. Fluid therapy is the basic therapeutic intervention to guarantee haemodynamic stability in the perioperative period and is mandatory in any anaesthetic episode [2]. Rational fluid administration is aimed to increase the pressure gradient for venous return, improving stroke volume (SV) and cardiac output (CO) in order to maintain and support tissue perfusion [1,3]. Hypovolaemia increases the risk of organ hypoperfusion, which may eventually lead to multiple organ failure or sepsis, whereas a large amount of fluid may induce pulmonary and peripheral edema and increase cardiac demand [3,4]. The administration of fluids titrated based on an assessment of individual preload dependency is recommended by several guidelines [1,3], and it has been shown to improve morbidity and mortality during surgery and ICU in people [5,6,7]. In veterinary medicine, the use of standard fixed fluid rates is still very popular [8]; however, this is unlikely to be appropriate for all cases because they could have different fluid needs [7].

Mean arterial blood pressure (MAP) and heart rate (HR) are the main clinical parameters used to assess the haemodynamic function and guide fluid therapy in animals (Davis et al., 2013). However, it has been shown that these parameters can remain in a physiological range even with variations up to 30% of total blood volume, and therefore they should not be considered adequate to guide fluid therapy, considering the risk of delayed response [9]. Pulmonary artery occlusion pressure and central venous pressure (CVP) are the more advanced static haemodynamic indices used to assess the response to fluid administration; however, their accuracy has been questioned because of the low sensitivity and specificity [10,11,12]. In dogs with sepsis, the CVP increase after a fluid bolus administration was not correlated with an increase in CO and MAP [13].

Fluid challenge (FC) is a dynamic test used to evaluate the preload reserve. It consists of the administration of a limited amount of IV fluids in a short period of time, in order to assess the cardiac response to an increase in intravascular volume (fluid responsiveness, FR) [14,15,16]. If the ventricles are functioning on the ascending portion of the Frank–Starling curve (preload dependency), then an improvement in the SV and its derivatives should be observed (fluid response). If the haemodynamic parameters remain unchanged or even deteriorate, then it is a clear indication that the patient would not benefit from further fluid administration (preload independency) [16,17,18]. The volume of the FC is an important aspect for the accuracy and the safety of the FC test. Where too small of a FC volume could be insufficient to stress out a preload dependency, a larger one may even lead to an excessive fluid administration and promote tissue edema [15,19,20]. The small volume FC has been investigated in people undergoing surgery [20], and they proved that a total of 100 mL is adequate to detect FR compared to the more classical 250 mL.

Recently, dynamic preload parameters have been shown to predict FR in mechanically ventilated patients [21,22]. Intermittent positive-pressure ventilation (IPPV) induces a cyclic reduction in the left ventricular preload, mainly by decreasing venous return. This change in preload throughout the respiratory cycle becomes even more significant during hypovolemia. Thus, these cyclic preload changes during respiration may result in SV, pulse pressure, and peripheral perfusion variations [16]. Dynamic preload parameters derived from the cyclic variation of the pressure waveform, such as the stroke volume variation (SVV), systolic pressure variation (SPV), and pulse pressure variation (PPV), or peripheral perfusion, such as the plethysmographic variability index (PVI), perform better than traditional static indices for predicting FR [17,18,23].

Studies in dogs confirmed the validity of dynamic indices for predicting FR in experimental conditions of hypovolemia due to haemorrhage [24,25] or during the perioperative period [12,26,27,28] using a FC volume between 10 and 20 mL/kg. Moreover Celeita-Rodriguez et al. [28] proved that PPV and PVI have a superior diagnostic accuracy compared to the other dynamic indices due to a narrower grey zone and a smaller number of cases within it. Nevertheless, a small-volume FC may be indicated in veterinary cases in order also to increase the safety of the procedure reducing the risk of fluid overload during surgery [20], as also reported by the AAHA/AAFP fluid-therapy guidelines for dogs and cats [8]. Data regarding the incidence of FR and the performance of the dynamic preload indices to a FC < 10 mL/kg in normotensive dogs are missing in the current veterinary literature. This information could be useful to guide fluid therapy in clinical cases in order to decide for a more restrictive or liberal fluids rates.

This study was designed to evaluate the incidence of FR to a small-volume FC in a population of normotensive dogs under general anaesthesia. Moreover, the accuracy of the PPV, PVI, SPV, and SVV in predicting FR in such conditions was evaluated.

The hypotheses were that (1) a small-volume FC would be able to detect FR in a group of normotensive anaesthetised dogs and (2) of the different dynamic preload indices, the PPV and PVI would be the most accurate in predicting FR. To test these hypotheses, a small-volume FC was performed in dogs under isoflurane anaesthesia to assess cardiac index (CI) change as a confirmation or exclusion of FR and to evaluate dynamic preload parameters performance before and after FC.

## 2. Materials and Methods

### 2.1. Animals

After an owner’s informed and written consent was acquired, dogs of varying breed, age, body weight, and sex undergoing surgery were included in this prospective clinical study. The inclusion criteria were as follows: body weight > 6 kg, American Society of Anaesthesiologists (ASA) physical status 1 or 2, dogs undergoing orthopaedic procedures and soft tissue procedures on the abdomen, eyelids, and skin, and a normal pre-anaesthetic blood count and chemistry, including renal parameters. The exclusion criteria were as follows: thoracic surgical procedures, contra-indication for arterial catheterization and mechanical ventilation, major cardiovascular (i.e., heart murmurs and mitral valve disease) and respiratory diseases, arrhythmias, surgical procedures with an estimated time <30 min, patient–ventilator asynchrony, dogs under cardiovascular-supporting drugs (e.g., dobutamine, dopamine, and ephedrine) and need for anaesthetic or analgesic drugs other than robenacoxib, methadone, propofol, and isoflurane. Dogs that showed an invasive MAP below 55 mmHg under a stable plane of anaesthesia, before the administration of the FC, were also excluded.

### 2.2. Anaesthetic Procedures

Before induction of anaesthesia, food and water were withheld for 12 and 2 h, respectively. At 30 min before the start of the procedure, the dogs were premedicated with an intramuscular (IM) injection of methadone (0.3 mg/kg; Semfortan^®^ 10 mg·mL^−1^; Dechra, Turin, Italy). Robenacoxib IM (1 mg/kg; Onsior^®^ 20 mg·mL^−1^; Elanco, Milan, Italy) was given alongside the premedication. After achieving an adequate level of sedation, the dogs were subcutaneously (SC) treated with 25 mg/kg of amoxicillin and clavulanic acid (Synulox 140/35 mg·mL^−1^; Zoetis, Morris County, NJ, USA), and a cephalic vein was catheterized for perioperative fluid administration (Ringer’s lactate solution 5 mL/kg/h) and drugs administration. All dogs were intravenously (IV) induced with propofol administered to effect (Proposure^®^ 10 mg/mL; Merial, Lyon, France), intubated using a cuffed orotracheal tube, and connected to the anaesthetic machine (Aliseo^®^ Anesthesia System, Datex Ohmeda, Helsinki, Finland) with a circle anaesthetic breathing circuit to maintain anaesthesia with isoflurane (Isoba^®^; Schering-Plough, Kenilworth, NJ, USA) in oxygen under mechanical ventilation with a piston-driven mechanical ventilator integrated in the anaesthesia workstation. Throughout the procedure, the end-tidal concentration of isoflurane (FE’Iso) varied between 1 and 1.3% according to individual needs. All dogs were mechanically ventilated in a volume-controlled mode with an expired tidal volume (VT) of 15 mL/kg, inspiration–expiration ratio (I:E) of 1:2, inspiratory pause of 10% of the inspiratory time, FiO_2_ > 0.8; and a respiratory rate (RR) adjusted to maintain the end-tidal carbon dioxide tension (PEtCO_2_) between 35 and 45 mm Hg [29]. Positive end-expiratory pressure was not administered (0 cmH_2_O), and the maximum peak airway pressure (Ppeak) was adjusted to 20 cmH_2_O for all animals. After induction, a 20–22 G catheter was placed (Seldinger technique) into the dorsal pedal artery for haemodynamic monitoring. The catheter was connected to a new precalibrated transducer with a dedicated saline-filled line and was zeroed at the right atrial level.

Basic physiological parameters (DATEX Ohmeda S/5, Helsinki, Finland) were recorded on a data sheet every 5 min and included the RR (breaths/min), HR (beats/min), PEtCO_2_ (mm Hg), (FE’Iso, %), capillary oxygen haemoglobin saturation (SpO_2_; %), invasive systolic (SAP; mm Hg), diastolic (DAP; mm Hg) and mean arterial pressure (MAP; mmHg), body temperature (T; °C), Ppeak (cmH_2_O), expired VT (mL), and respiratory system dynamic compliance (Crs; mL/cmH_2_O). Spirometry was assessed with the pitot tube of the anaesthesia monitor placed at the level of the endotracheal tube.

A dose of rescue analgesia (2 ug/kg of fentanyl) was administered whenever the HR, MAP, or RR increased >20% during surgical manipulation. At the end of each procedure, isoflurane administration was interrupted, and dogs were disconnected from the circuit when the SpO_2_ and PEtCO_2_ values were within the normal range. When the swallowing reflex returned, the orotracheal tube was removed.

### 2.3. Haemodynamic Monitoring

Relevant haemodynamic monitoring was performed using the pressure recording analytical method (PRAM; Most Care; Vytech) [30]. Heart rate, SAP, DAP, MAP, SPV (%), CO (L/min), PPV (%), and SVV (%) were continuously monitored based on the following formulas:PPV (%) = 100 × (PPmax − PPmin)/(PPmax + PPmin)/2)
SPV (%) = 100 × (SPmax − SPmin)/(SPmax + SPmin)/2
SVV (%) = 100 × (SV max − Svmin)/(SVmax + SVmin)/2
CO (L·min − 1) = SV × HR

The CI (L/min/m^2^) was computed by dividing CO by the body surface area (0.101 × (BW in kg)⅔; m^2^). The stroke volume index (SVI; mL/m^2^/beat) was computed by dividing the CI by the HR.

The number of pulse waveforms over which the PRAM measured the dynamic indexes was modified for each case based on the heart and respiratory rates of the dog in order to make sure that the indices were calculated within the same respiratory cycle.

Every 30 s, the monitor automatically collected data for all of the parameters, which were transferred to an Excel sheet for off-line evaluation.

The PVI was assessed using a Massimo rainbow SET^®^ pulse oximeter (Irvine, CA, USA), which noninvasively calculated this parameter by analysing the perfusion index (PI). The PI is the ratio of pulsatile to nonpulsatile blood flow through the peripheral capillary bed. The PVI reflects dynamic breath-to-breath changes in the PI and is automatically calculated using a specific algorithm (PImax − PImin)/PIma × 100 [23,31]. The pulse oximeter probe was clipped on the tongue.

### 2.4. Study Protocol

Before starting the surgical procedure, FC was performed after obtaining an adequate level of anaesthesia (absence of palpebral reflex, ventral–medial rotation of the eye, and loss of the jaw tone) and confirming via pressure/volume loop and capnography observation that there was no patient/ventilator asynchrony. Electrocardiography was evaluated before the start of the FC in order to detect any arrythmias that could affect the haemodynamic monitoring. In this case, dogs were excluded from the study.

Dogs with a MAP lower than 55 mmHg, and/or under cardiovascular supporting drugs at the time of the FC, were not included in the study. FC consisted of the infusion of 5 mL/kg of Ringer’s lactate solution within 5 min using (60 mL/kg/h) one or two peristaltic infusion pumps (LP 7700; AMPall Co. Ltd., Seul, Korea) [15,32]. The infusion pumps were tested for accuracy before each case and a standard 120 cm infusion set was used in all dogs. Arterial pressure signal adequacy was evaluated with square-wave testing and visual observation of the damping before each measurement. In case of under (more than two oscillation after the flush test) or over (none or only one oscillation after the flush test) damping, the dog was excluded from the study. The HR, SAP, MAP, DAP, CO, SV, PPV, SPV, SVV, and PVI were collected at baseline, 2 min before bolus administration (PRE), and 2 min after bolus administration (POST).

During the procedure, markers were assigned at the beginning and end of FC to identify the PRE and POST CI, HR, SAP, MAP, DAP, PPV, SPV, and SVV. For each study time point, the cardiovascular parameter values were calculated as the mean of three consecutive recordings collected by the software every 30 s.

The PVI was also manually collected at each study time point (PRE and POST).

### 2.5. Statistics

All data were analysed using MedCalc software (v. 15.6.1; Medcalc, Ostend, Belgium).

Using preliminary data from the first twenty cases, a power calculation was performed for a 15% difference in the CI after FC. Using freely available software (G*Power v. 3.0.10; University of Düsseldorf, Düsseldorf, Germany), the power calculation was performed using a two-tailed *t*-test with a power of 0.95, an alpha error of 0.05, and effect size of 0.40 [33]. This analysis suggested that 78 dogs would be sufficient to detect a significant difference in the CI after FC. The preliminary cases were included in the population of the study.

Data results were normally distributed based on the Kolmogorov–Smirnov test and are expressed as mean ± standard deviation (SD) for the PRE and POST time points. Dogs that showed a CI increase ≥15% POST compared with PRE were considered responders (Rs), whereas the rest of the population was classified as nonresponders (NRs). Differences between Rs and NRs were assessed by two-tailed Student’s *t*-tests. Comparisons within groups were assessed using Student’s paired *t*-tests. Receiver operating characteristic (ROC) curves were used to evaluate the capacity of the PPV, PVI, SPV, SVV, and MAP to predict FR after FC. The areas under the ROC curve (AUCs) were determined as previously described [34]. The best cut-off values to discriminate between the Rs and NRs were defined by a point on the ROC curve determined by the maximum Youden index value (sensitivity + specificity − 1) [35]. The AUCs were compared using the Hanley–McNeil test (AUC, 0.5, a test with no predictability; AUC, 0.6–0.69, a test with poor predictability; AUC, 0.7–0.79, a fair test; AUC, 0.8–0.89, a test with good predictability; AUC, 0.9–0.99, an excellent test; and AUC, 1.0, a perfect test with the best predictability).

The grey-zone approach [36] was used to determine each parameter’s inconclusive range using the two-steps approach as previously reported in the literature [28,37]. The percentage of individuals of the original study population that fell into the grey zone was calculated as an estimate of the diagnostic accuracy. A Spearman correlation test was used to evaluate the correlations between the PVI and PPV, and SPV and SVV. A *p*-value < 0.05 was considered statistically significant.

## 3. Results

Of the 92 recorded cases, five were excluded because they required a modification of the anaesthetic protocol during the procedure, and seven were excluded because they were not being completely permissive to the mechanical ventilation at the time of measurement. Accordingly, data analysis was performed for 80 dogs, which had a mean age of 45.1 ± 19.5 mo and body weight of 24.1 ± 15.9 kg. All dogs were normotensive before the induction of anaesthesia and the average MAP value for the entire population was 97.5 ± 12.3 mmHg. The FE’Iso before and after the FC was similar (PRE = 1.18 ± 0.12% and POST = 1.21 ± 0.14%). Table 1 shows the cardiovascular and respiratory parameter values recorded from the entire study population before (PRE) and after (POST) FC administration.

The CI and SVI were higher POST compared with PRE. Moreover, the PPV and PVI decreased POST compared with PRE (Table 1). All of the other parameters were similar for the PRE and POST time points. Of the 80 dogs, 50 (62.5%) had an increased CI ≥ 15% (+38.1 ± 20.4%) POST FC and therefore, were classified as Rs, whereas the remaining 30 (37.5%) where considered NRs (−4.9 ± 10.4%). Body weight and age were similar between Rs (28.3 ± 17.67 kg and 42.12 ± 27.91 mo) and NRs (17.73 ± 9.26 kg and 38.38 ± 19.73 mo). Table 2 shows the mean ± SD of PRE and POST cardiovascular parameters divided between the R and NR groups.

The PRE CI and SVI were lower in the Rs compared with the NRs. The POST CI and SVI increased significantly in the R group, whereas these parameters did not vary in the NR group (Table 2). The PRE PVI and PPV were higher in the R group compared with the NR group. In the R group, the POST PVI and PPV were significantly lower than the PRE PVI and PPV. The PRE SPV and SVV were higher in the Rs compared with the NRs (Figure 1).

In Table 3, the cut-off values, sensitivities, specificities, AUCs, grey-zone limits, and number of grey-zone cases are reported for each dynamic variable and the MAP.

All of the dynamic parameters were predictive of FR (AUC > 0.5). The PPV and PVI were the most predictive of FR with similar AUCs > 0.9 (*p* = 0.96) and the narrowest grey-zone ranges. The SPV and SVV had significantly lower AUC values (*p* < 0.05) and wider grey-zone ranges compared with the PPV and PVI. The MAP was not predictive of FR (*p* = 0.378). The PVI showed a strong (r = 0.78) correlation with the PPV and a moderate correlation with the SPV and SVV (r = 0.58 and 0.42, respectively).

## 4. Discussion

The results of this study proved that, despite normal values of MAP and HR, up to 62.5% of dogs under general anaesthesia can be preload dependent when tested with a small-volume FC. This condition can be predicted with the evaluation of the dynamic preload parameters, with the PPV and PVI being more accurate than the SPV and SVV (Table 3). Our data confirmed that MAP, in the conditions evaluated in this study, is a poor predictor of FR and that preload dependency may not be associated with hypotension. The HR was not affected by the administration of the FC both in responder and nonresponder dogs, and this could be related to the fact that animals were normotensive at the time of FC administration.

Many factors may influence the haemodynamic condition of a dog under general anaesthesia, including the anaesthetic drugs used, body positioning, body habitus, and surgical and anaesthetic techniques [38,39]. In our dogs, propofol and isoflurane administration may have induced vasodilatation, which could mildly decrease the stressed venous blood volume and consequently, lead to a decrease in the SV and CI, which would move the heart to the steep portion of the Frank–Starling curve. This condition could have been further aggravated by the use of IPPV [4,40]. Suboptimal values of SV and CI are most likely responsible for suboptimal oxygen delivery (DO_2_) [1,38], and several human studies have suggested that DO_2_ optimization during surgery may improve patient outcomes [21,41,42]. Thus, we can speculate that monitoring of the dynamic preload indices during anaesthesia may give an earlier prediction of haemodynamic instability, even in cases not associated with classical parameter (e.g., MAP and HR) alterations and may be a valid tool to guide fluid therapy [1]. Further canine studies, in different physiological or pathological conditions, should clarify whether early haemodynamic optimisation in these cases improves the outcome and reduces postoperative complication risks as proven in people [43].

The small FC volume used in this study is in the range recently suggested for dogs under general anaesthesia [8] and can be considered less aggressive for haemodynamic stable clinical cases to avoid the risk of overhydration especially in cases in which the FC should be repeated during surgery. Other similar veterinary studies have used larger volumes for the FC: Sano et al. [12] 10 mL/kg of hydroxyethyl starch administered in 13 min; Fantoni et al. [26] 15 mL/kg of Ringer’s lactate administered in 15 min; Celeita-Rodriguez et al. [28] 20 mL/kg of Ringer’s lactate in 15 min; Drozdzynska et al. [27] 0 mL/kg of sodium lactate solution administered in 10 min; Rabozzi & Franci [44] 3 mL/kg of Hartmann’s solution administered in 1 min; and Bucci et al. [45] 5 mL/kg of Ringer’s lactate solution in 1 min. If we exclude the last three studies characterised by completely different settings and protocols, the rest of the studies gave 58% [12], 76% [26], and 100% [28] rate of responders. The rate of responders found in this study (62.5%), fits in between the previous results, despite the smaller volume of the FC, but we cannot exclude that the relatively small volume of FC used could have provided false negative cases, and future studies are required to address the most adequate volume to perform a FC in dogs.

However, human studies have demonstrated that small-volume (100 mL) FC is similarly effective compared with large-volume (500 mL) FC [20] in detecting FR and that a minimum FC volume of 4 mL/kg of actual body weight is required to test heart responsiveness [15,46]. Small-volume FC is popular in humans due also to the safety considerations for the patients [37]. The rate of the fluid administration plays also an important role in the impact of the FC, in particular, considering a constant volume, a rapid intravascular fluid bolus will be more effective in increasing venous return and consequently the right ventricle end-diastolic volume, as compared to a slower rate of infusion [46]. The pharmacodynamic analysis of an FC showed that the maximal change in CO is observed 1 minute after the end of the infusion of a crystalloid solution and dissipated over 10 min period. Accordingly, infusion times shorter than 10 min are recommended [32]. In this study we used an infusion rate of 60 mL/kg/h and we cannot exclude that higher rates could have increase the incidence of responsiveness to the FC. Future studies should try to address this important aspect of the FC.

The PRAM is an uncalibrated pulse contour technique that has been recently validated in dogs [30] and proved to have the best trending ability among the other uncalibrated pulse contour techniques in this species, under stable haemodynamic conditions. The PRAM estimates SV and other haemodynamic parameters based on arterial pulse waveform analysis [47]. SV (mL) is estimated by measuring the area under the systolic portion of an arterial pressure curve and the variable Z. The variable Z represents all of the possible factors responsible for the dynamic impedance of the cardiovascular system, which oppose pressure wave propagation in arterial beds, and is computed from the pressure curve through a proprietary algorithm applied to each beat [30,47]. The PRAM technique has been reported in the literature for the assessment of the FC in infants undergoing ventricular septal defect [48]. Franchi et al. [49] demonstrated that PRAM technology was able to detect rapid variation of CO in septic patients receiving norepinephrine therapy. Other studies, on the contrary, demonstrated a poor accuracy of the PRAM to detect variations of CO in people receiving a FC [50,51]. Uncalibrated pulse contour technologies have been traditionally considered less reliable than the calibrated techniques in detecting a rapid variation of the CO (e.g., FC, vasopressor, and sepsis) [52]. However, we should consider that PRAM is different from the other uncalibrated pulse contour methods, because the arterial impedance that is required for calculation of SV is estimated from perturbations of the arterial pressure waveform and not derived from internal nomograms based on demographic parameters [53]. The aim of our study was not to test the accuracy of PRAM in detecting rapid variation of CO; however, considering the absence of an alternative direct method to assess CO in the protocol of our study, we cannot exclude that in some cases the PRAM algorithm could have failed to detect the volume change after the FC.

The results of this study confirmed that the PPV and PVI are excellent (AUC 0.9–0.99) dynamic indices for predicting FR in dogs compared to the SPV and SVV which can be considered fair (AUC, 0.7–0.79) based on the Hanley–McNeil test. Compared with the other study parameters, the PPV and PVI also had the smallest grey–zone ranges and lowest number of cases within those ranges (Table 3). If we compare our results with that of Celeita-Rodriguez et al. [28], in which the same statistical approach has been used, the cut-off values, the width of the grey zone and the number of cases in this zone, for PPV and PVI were smaller in our study, indicating a greater accuracy, probably related to the larger population of dogs. However, we should consider that Celeita-Rodriguez et al. [28] used 20 mL/kg of FC. The PPV and PVI cut-off values are very similar and considering the strong correlation between them, these values may be considered interchangeable. The PVI has the adjunctive value of not requiring an invasive arterial line and thus, is less invasive and accessible to less-equipped clinics. However, in cases with pathological (e.g., haemorrhage and sepsis) or iatrogenic (e.g., pain, hypothermia and the use of alpha-2 agonists) perfusion reduction, there are limitations to the variation of the peripheral perfusion that should be considered with the PVI. The PVI requires a minimum of 1% of the PI to produce a reliable value, which may be the main limitation of this technology [23].

The SPV had a lower accuracy in predicting FR compared to the PPV and PVI (Table 3; Figure 1); however, considering its high AUC, sensitivity and specificity and narrow grey-zone range, this parameter can still be valid for the assessment of FR. The SPV cut-off value (>4.1%) is very close to that reported by Rabozzi and Franci [44] (>4.5%). However, given that the SPV is a pressure-derived parameter that is always monitored by systems that measure the PPV, clinicians should preferentially use the latter parameter because of its greater accuracy.

Of the examined parameters, the SVV had the lowest accuracy for predicting FR (Table 3; Figure 1), which is in agreement with previous human and veterinary studies [17,22,28]. Our SVV cut-off value of >14.7% is higher than that reported by Endo et al. [54] (>11%) and Celeita-Rodriguetz et al. [28] (>10%). We may speculate that the difference in SVV cut-off thresholds could be related to the use of a calibrated methods to measure real time SV (PICCOTM) in the two studies mentioned above.

The fact that the variations in CI and the estimation of the dynamic preload indexes were performed with the same monitor, could have add a potential bias to the study and should be consider a limitation. Moreover, another limitation of the study is that Ppeak delivered to the animals was not standardised as suggested by the literature [1], since it is an important determinant of the heart–lung interactions. The damping coefficient was not calculated in this study, and we cannot exclude that, in some cases, the quality of the arterial trace was not optimal and may have influenced the accuracy of the measurements.

## 5. Conclusions

Preload dependency is a condition that is detectable in 62.5% of normotensive dogs undergoing inhalational general anaesthesia with 5 mL/kg of FC. In such conditions, PPV and PVI are the most accurate dynamic preload indices for predicting FR and can be used interchangeably, given the management of the case and the equipment available. The SPV and SVV also have a good predictability of FR predictiveness; however, their accuracies are lower than those of the PPV and PVI. Further studies are required to evaluate whether the small volume of the FC used in this study could have underestimated the real incidence of FR and the impact of dynamic preload parameters monitoring on the management of intraoperative fluid therapy in dogs.

## Figures and Tables

**Figure 1 vetsci-08-00026-f001:**
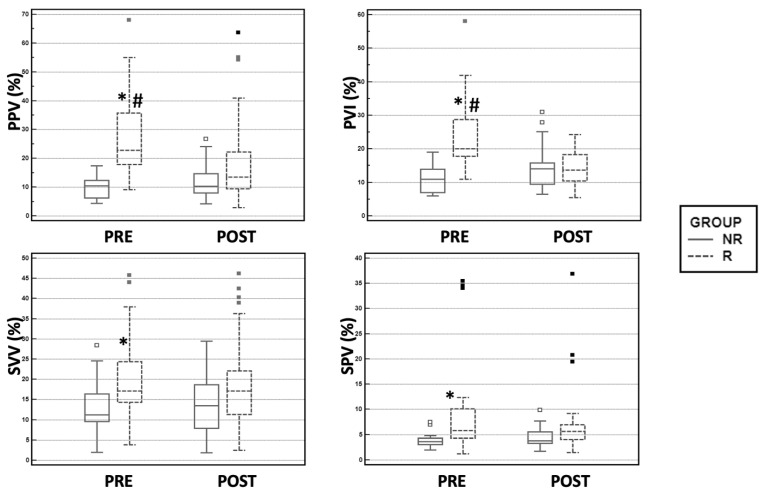
Box plot representation of the pulse pressure variation (PPV), plethysmographic variability index (PVI), stroke volume variation (SVV), and systolic pressure variation (SPV) before (PRE) and 2 min after (POST) fluid challenge administration of 5 mL/kg of Ringer’s lactate within 5 min. Based on the POST cardiac index (CI) variation, dogs were classified as responders (Rs; CI ≥ 15%) or nonresponders (NRs). The line within the box represents the median value. The upper and lower limits of the boxes represent the 25th and 75th percentiles of the data, respectively. The whiskers extend from the minimum to the maximum value. * *p* < 0.05 compared to NR; # *p* < 0.05 compared to POST within the same group.

**Table 1 vetsci-08-00026-t001:** Mean and 95% confidential intervals of cardiovascular and respiratory parameters collected from 80 dogs under general anaesthesia before (PRE) and 2 min after (POST) a fluid challenge administration of 5 mL/kg of Ringer’s lactate within 5 min.

Parameters	PRE	POST
HR (beats/min)	89.2 (85.1–93.4)	90.6 (86.5–94.9)
SAP (mmHg)	103.3 (98.1–108.6)	109.5 (104.2–114.7)
DAP (mmHg)	64.7 (60.6–68.8)	67.1 (63.3–71.0)
MAP (mmHg)	77.1 (72.9–81.2)	79.9 (76.1–83.6)
CI (L/min/m^2^)	3.5 (3.1–4.0)	4.1 (3.7–4.5) *
SVI (mL/beat/m^2^)	41.7 (36.9–46.6)	46.7 (42.3–51.1) *
PPV (%)	20.9 (17.8–24.0)	15.8 (13.3–18.3) *
PVI (%)	18.9 (16.5–21.3)	14.2 (12.9–15.5) *
SPV (%)	7.3 (4.5–10.2)	6.5 (4.5–8.5)
SVV (%)	17.2 (15.2–19.2)	16.7 (14.5–18.9)
Ppeak (cmH_2_O)	10.6 (9.8–11.4)	10.6 (10.0–11.3)
PEtCO_2_ (mmHg)	41.8 (33.2–46.3)	42.2 (34.3–44.3)
SpO_2_ (%)	97.5 (95.2–98.5)	97.2 (95.3–98.2)
Crs (mL/cmH_2_O/kg)	1.5 ± 0.7 (1.2–1.8)	1.6 (1.2–1.9)

HR = heart rate; SAP = systolic arterial pressure; DAP = diastolic arterial pressure; MAP = mean arterial pressure; CI = cardiac index; SVI = stroke volume index; PPV = pulse pressure variation; PVI = plethysmographic variability index; SPV = systolic pressure variation; SVV = stroke volume variation; Ppeak = peak airway pressure; PetCO_2_ = end-tidal carbon dioxide partial pressure; SpO_2_ = capillary oxygen haemoglobin saturation; and Crs = dynamic compliance of the respiratory system indexed for body weight. * *p* < 0.05 compared to PRE.

**Table 2 vetsci-08-00026-t002:** Mean ± standard deviation of cardiovascular parameters recorded from 80 dogs under general anaesthesia and mechanically ventilated before (PRE) and after (POST) a fluid challenge of 5 mL/kg of Ringer’s lactate solution within 5 min. Responders (Rs) were dogs that had a CI increase ≥ 15% POST compared with PRE, whereas the rest of the population was classified as nonresponders (NRs).

Parameters	Time	Responders (*n* = 50)	Nonnresponders (*n* = 30)
HR (beats/min)	PRE	89.8 ± 20.9	88.4 ± 13.9
	POST	91.2 ± 18.7	90.1 ± 19.1
CI (L·min/m^2^)	PRE	2.86 ± 1.09	4.8 ± 2.4 *
	POST	3.9 ± 1.4 ^#^	4.4 ± 2.1
SVI (L/min/m^2^)	PRE	34.3 ± 16.2	53.6 ± 23.8 *
	POST	44.7 ± 18.5 ^#^	49.9 ± 21.1
SAP (mmHg)	PRE	99.1 ± 20.9	110.3 ± 25.1
	POST	109.5 ± 22.7	109.4 ± 24.9
DAP (mmHg)	PRE	63.2 ± 17.1	67.1 ± 19.8
	POST	67.9 ± 15.6	65.9 ± 19.2
MAP (mmHg)	PRE	75.8 ± 18.2	79.1 ± 19.1
	POST	80.7 ± 16.2	78.5 ± 17.6
PVI (%)	PRE	23.6 ± 9.7	10.7 ± 3.5 *
	POST	14.3 ± 5.2 ^#^	14.1 ± 6.1
PPV (%)	PRE	27.6 ± 13.5	9.7 ± 3.5 *
	POST	18.1 ± 12.8 ^#^	12.1 ± 5.6
SPV (%)	PRE	10.1 ± 10.6	3.9 ± 1.4 *
	POST	7.8 ± 14.1	4.6 ± 2.1
SVV (%)	PRE	19.4 ± 9.3	13.5 ± 6.4 *
	POST	18.8 ± 10.7	13.2 ± 6.9

HR = heart rate; SAP = systolic arterial pressure; DAP = diastolic arterial pressure; MAP = mean arterial pressure; CI = cardiac index; SVI = stroke volume index; PPV = pulse pressure variation; PVI = plethysmographic variability index; SPV = systolic pressure variation; SVV = stroke volume variation; * *p* < 0.05 compared to R; ^#^
*p* < 0.05 compared to PRE within the same group.

**Table 3 vetsci-08-00026-t003:** Predictiveness of different dynamic indices and MAP in 80 dogs receiving fluid challenge under general anaesthesia. The best cut-off values were determined based on the Youden index. Cut-off values delimiting the grey zones were defined by values associated with 90% sensitivity and 90% specificity.

Dynamic Index	Cut-Off	Sensitivity (% (95% CI))	Specificity (% (95% CI))	AUC (95% CI)	*p*	Grey Zone (%)	Grey Zone Cases (No. (%))
PPV	>13.8%	98.0 (89.3–99.7)	93.3 (77.9–99.0)	0.979 (0.919–0.997)	<0.0001	13.5–15.5	6 (7.5)
PVI	>14%	93.3 (81.7–98.5)	92.3 (74.8–98.8)	0.956 (0.878–0.990)	<0.0001	13–15	10 (12.5)
SPV	>4.1%	85.7 (63.6–96.8)	75.0 (47.6–92.6)	0.793 (0.628–0.908)	<0.0001	3.1–5.1	23 (28.75)
SVV	>14.7%	72.0 (57.5–83.8)	73.3 (54.1–87.7)	0.729 (0.618–0.822)	<0.0001	12.7–15.3	25 (31.2)
MAP	>67 mmHg	41.67 (27.6–56.8)	80 (61.4–92.3)	0.560 (0.443–0.672)	0.3788	<98–>56	61 (76.2)

PPV = pulse pressure variation; PVI = plethysmographic variability index; SPV = systolic pressure variation; SVV = stroke volume variation; MAP = mean arterial pressure; AUC = area under the curve.

## Data Availability

The data presented in this study are available on request from the corresponding author. The data are not publicly available due to institutional privacy policy.

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
