# Peer review of "Intraoperative Assessment of Fluid Responsiveness in Normotensive Dogs under Isoflurane Anaesthesia"

_vetsci, 2021, doi:10.3390/vetsci8020026_

Round 1

Reviewer 1 Report

Reviewer’s comments:

Intraoperative Assessment of Fluid Responsiveness in Normotensive Dogs Under Isoflurane Anaesthesia

In general

This is very well designed and described study. There is a very good overview of the literature in the introduction. The aim of the study is clearly defined. The used technique is well described. There is in my opinion only one main aspect, which is missing for me. There are no data available in this study of the preclinical examination and also there are no informations about the blood count and blood chemistry. This could be really important in patients with abnormal blood counts and abnormal fluid and electrolytes. The renal values should also be written down, because this is also important in relation to the blood pressure. This is also a point I would like to have is a pre anaesthetic blood pressure measurement. This is especially important because the title of the manuscript is:… in normotensive dogs. The results are shown quiet nicely in the diagrams. The discussion is excellent written with a good description of the literature. In my opinion it would be perfect to fit in the missing data or to argue why they are missing. So after a minor revision the manuscript should be published. 

Abstract

-

Introduction

The introduction is an excellent overview of the literature with a good defined hypothesis of the current study.

Material and methods:

Animals

Mean arterial pressure below 55mmHg: Is that before or under general anaesthesia ? measured with what kind of device?

There is a pre-anaesthetic examination missing and a calculation of the clinical hydration status and a blood pressure measurement before ?

Was there a normal blood count, because volume and fluid demand would be different in patients with not a normal hematocrit.

Aneasthetic procedures

If there is given Onsior with the premedication, is there a pre-anaesthetic blood chemistry for renal values?

Why is the oxygen content in these patients with a FiO2 > 0,8 so high?

The arterial catheter: with which technique was it introduced (Seldinger?)

Haemodynamic monitoring:

-

Study protocol

Adequate level of anesthesia: how is that defined? Which parameters were obtained to quantify that?

Statistics:

Normally distributed…. This sentence is not really correct, because the Kolmogorov test (quiet old test) is used to evaluate if the data are normally distributed and then they are expressed as mean…

All data were normally distributed?

Discussion

-

Reviewer 2 Report

Dear Authors,

Thank you for this interesting study that aims to assess changes in dynamic indices of stroke volume following administration of a fluid challenge to normotensive anaesthetised dogs. I think this is an original study that will be of great interest to veterinary clinicians, particularly those working in anaesthesia and critical care.

I do have some comments that I would appreciate the authors consider and respond to.

Overall grammar:  Some editing is required for entire manuscript (particularly introduction) required for relatively minor but frequent English language errors. I have not listed each error as they appear too frequently for this to fall within the scope of a peer reviewer.

Introduction

“Rational fluid administration is the main therapeutic intervention that guarantees haemodynamic stability in anaesthetized patients”:

  • Is there any evidence to support the claim that fluid administration “guarantees” haemodynamic stability? The reference provided supports the last part of the statement that fluid therapy is “mandatory” during anaesthesia but not the first part of this claim… There will be cases where haemodynamically unstable animals are not responsive to fluids during anaesthesia (in fact that is why this paper has been written), so I don’t think you can say that fluid administration guarantees stability.

  • Does this sentence refer to veterinary patients, or to people? It is not clear from the sentence, although the reference is a veterinary guidelines.

  • It is worth adding an explanation here of what is meant by haemodynamic stability. In this study, the authors use cardiac index as the “reference standard” for fluid responsiveness, and therefore haemodynamic stability. So, it is important to “tie” together the concepts of haemodynamic stability, fluid responsiveness, stroke volume, and cardiac index at the beginning of the introduction

No need to write “human patients”. By definition, a “patient” is a person. I recommend revising to “people”. Likewise, ideally “patient” should not be used to refer to an animal, better to write “animals”.

Check for consistent usage of acronyms, and their definition, throughout the manuscript. These are sometimes used prior to their definition; for example, FR is used in the second paragraph of the introduction for the first time but is not defined as “fluid responsiveness” until paragraph 3. Then, in the last sentence of the introduction the phrase is spelled out fully rather than the acronym now being used.

Hypotheses: what do the authors mean by “in certain percentage” of dogs? Consider revising to “in a group of normotensive anaesthetised dogs”. Or, if the authors mean that they hypothesised a certain number of these dogs would be fluid responsive, then the actual number hypothesised should be stated (e.g. “in 50% of dogs”).

Methods

When they have stated in the introduction, that improvement in stroke volume is the best indicator of fluid responsiveness, why do the authors use cardiac index, rather than stroke volume index, as the outcome measure of fluid responsiveness?

Given that mechanical ventilation technique has potential to impact the outcomes of this study, more detail on this is needed:

  • Please provide the make and mode of the anaesthetic workstation used for ventilation

  • Can the authors provide more information on the ventilation mode used? I presume this was mandatory volume-controlled ventilation. There should be some mention here (first paragraph of “anaesthetic procedures section”) that the peak inspiratory pressure was not the same for all animals. Was a maximum PIP set?

  • I assume that PEEP was set to zero. Again, this is important to note here.

  • How did the anaesthetic workstation measure delivered tidal volume, and was this recorded? Given later mention of pressure-volume and flow-volume loops, and compliance values; I presume there was in-built spirometry but this should be described here.

Was ECG monitoring performed to identify any arrythmias that might have affected SVV?*

“A dose of…fentanyl… was expected..” Consider revising the phrase “was expected”. “Was administered” is more appropriate.

Was the accuracy of the infusion pumps used to administer the fluid challenge checked prior to use? What type/length of extension tubing was used with the pumps, and was this the same for all dogs?

Results

Figure 1:

  • There should be an explanation of what data was used to create the box plots in the legend (i.e., median and quartiles).

  • Is it necessary to have the data for PPV, PVI, SVV, and PSV in both Table 2 and Figure 1? If this data was normally distributed and the mean and SD already reported in table 1, I am not sure of the need for the figure. Consider removing figure or removing this data from the table.

Discussion

Paragraph 3 sentences 2 and 3. This list of previous studies produces quite a long and confusing sentence. Consider revising to remove mention of the last three studies that the authors exclude anyway in the next sentence and producing one sentence that lists each study and gives the % rate of responders immediately after mention of the study. For example: “Sano et al. (2018) report a 58% response rate following 10mL/kg of HES over 13 minutes; Fantoni et al (2017) report a 76% reponse to ….”

Paragraph 4. If it has been shown that the maximal change in CO following a FC is observed 1 minute after the end of infusion, why did the authors choose to collect data at 2 minutes following the end of infusion in the present study? Is it possible that peak effect was missed in some dogs?

Reviewer 3 Report

Reviewer comments for manuscript ID vet sci-1074705 entitled ‘Intraoperative Assessment of Fluid Responsiveness in Normotensive Dogs Under Isoflurane Anaesthesia’

General comments

A very important and often neglected aspect in canine clinical anaesthesia has been thoroughly researched in this study. Data from 80 dogs in clinical settings provides robustness to the study. Moreover, its applicability even in less equipped clinical settings is the hallmark of this study. I am really impressed with the design and statistical treatment of the data. The study should be able to provide anaesthesiologists a reliable guideline for fluid challenge and therapy in clinical conditions in canines. I congratulate the authors for a thorough research and a well written manuscript. Each aspect of the manuscript is very nicely written and explained. The discussion is nicely presented with a clear take home message and even highlighting of limitations of the study.

I once again congratulate the authors for such as professional approach in the design, execution, and communication of such an important but often neglected aspect of canine anaesthesia.

I could not find any mistakes in the manuscript despite taking a bit longer time to review this manuscript. I recommend the publication of the manuscript.

Specific comments

Introduction: Refer para 5 -Please reframeAt the moment when this manuscript was draw up,’ as ‘At the time of writing this paper’

Discussion: Refer para 1 – Please change ’preload’ to ‘preloaded’
